# Neuroprotective Peptides and New Strategies for Ischemic Stroke Drug Discoveries

**DOI:** 10.3390/genes14050953

**Published:** 2023-04-22

**Authors:** Lyudmila V. Dergunova, Ivan B. Filippenkov, Svetlana A. Limborska, Nikolay F. Myasoedov

**Affiliations:** Institute of Molecular Genetics, National Research Center “Kurchatov Institute”, Kurchatov Sq. 2, 123182 Moscow, Russia

**Keywords:** ischemic stroke, peptides, peptide regulation, transcriptomic analysis, RNA-Seq, brain, neuroprotection

## Abstract

Ischemic stroke continues to be one of the leading causes of death and disability in the adult population worldwide. The currently used pharmacological methods for the treatment of ischemic stroke are not effective enough and require the search for new tools and approaches to identify therapeutic targets and potential neuroprotectors. Today, in the development of neuroprotective drugs for the treatment of stroke, special attention is paid to peptides. Namely, peptide action is aimed at blocking the cascade of pathological processes caused by a decrease in blood flow to the brain tissues. Different groups of peptides have therapeutic potential in ischemia. Among them are small interfering peptides that block protein–protein interactions, cationic arginine-rich peptides with a combination of various neuroprotective properties, shuttle peptides that ensure the permeability of neuroprotectors through the blood–brain barrier, and synthetic peptides that mimic natural regulatory peptides and hormones. In this review, we consider the latest achievements and trends in the development of new biologically active peptides, as well as the role of transcriptomic analysis in identifying the molecular mechanisms of action of potential drugs aimed at the treatment of ischemic stroke.

## 1. Introduction

Ischemic stroke (IS) continues to be one of the leading causes of death and disability in the adult population worldwide. In the early stages after the onset of an ischemic attack, effective treatment for stroke still includes the use of thrombolytic agents or mechanical thrombectomy [1,2,3,4]. Timely successful reperfusion is the most effective treatment for patients with acute IS, but vascular recanalization can lead to ischemia-reperfusion (IR) injury, and thrombolytic drugs have no effect on protecting or reversing neuronal damage [2,5,6]. The treatment of IS is not limited to reperfusion therapy. Drug therapy includes antiplatelet agents, anticoagulants, antioxidants, antihypertensives, anti-excitotoxic calcium-stabilizing drugs, and other drugs [7,8,9,10]. Despite the versatility and diversity of the drugs used, the currently existing pharmacological methods for the treatment of IS are not effective enough and require further study of the molecular basis of ischemic damage, the development of new therapeutic agents, and approaches to identify potential neuroprotectors. The use of neuroprotectors to protect nerve cells from damage and death and to improve the activity of the nervous system is one of the main directions in the pathogenetic treatment of many neuropathologies. Among them are encephalopathies of various origins, neurodegenerative diseases, the consequences of traumatic brain injuries, chronic cerebrovascular accidents in the elderly, and acute cerebral ischemia. Neuroprotectors have heterogeneous chemical structures and different mechanisms of action. Herbal drugs, antioxidants and vitamins, calcium channel blockers, and agents that improve cerebral metabolism and affect neurodegeneration have a neuroprotective effect [11,12,13,14,15]. Promising candidates for neuroprotector roles are agents with pleiotropic activity, which have antioxidant, anti-inflammatory, regenerative, antiplatelet, and antiapoptotic properties. Peptide drugs are used as neuroprotectors to treat many diseases [16,17,18].

Peptides are a unique class of pharmaceutical compounds with high potency, clear specificity, low immunogenicity, biocompatibility, mild action, and no side effects. Efforts and achievements in the discovery, production, and modification of peptide drugs in recent years, as well as their current application in various pathologies, have been summarized in several reviews [19,20,21,22,23]. In the search for new therapeutic agents for stroke prevention, peptides are of great interest. Today, in the development of neuroprotective drugs for the treatment of stroke, special attention is paid to peptides that act in several key areas: (1) in blocking the cascade of pathological processes caused by the restriction of blood flow to the brain tissues, (2) in preventing repeated violations of the blood supply to the brain and complications that aggravate the course of the disease, and (3) in restoring nerve tissue and brain function after ischemia [24,25,26].

A significant contribution to the development of new neuroprotective drugs is made by transcriptome studies based on genome-wide sequencing of cellular RNAs. This approach makes it possible to detail the changes in gene expression during the development of IS in order to identify the signaling pathways involved in the molecular mechanisms of damage. As a result, genome-wide analysis makes it possible to detect potential therapeutic targets, the impact of which can serve as a new approach for the treatment of the pathological process. The use of the transcriptomic approach has made it possible to study the mechanisms of action of many natural medicinal compounds and peptide drugs successfully used for the treatment of IS [27,28,29].

Current problems in the development of neuroprotective strategies, shortcomings in preclinical modeling, lessons from earlier approaches in clinical trials, and ways to overcome these problems have been discussed in previous reviews [8,26,30,31]. Previously, we have described, in detail, medical technologies for the treatment of IS using drugs based on natural regulatory peptides [27,28,32]. Additionally, the effects of several peptides on brain impairment conditions, including NPY, substance P, galanin, nocistatin, and sectroneurin, have also been described in reviews [33,34,35]. In this review, we consider the latest achievements and trends in the development of new biologically active peptides, as well as the role of genome-wide transcriptome studies in identifying the molecular mechanisms of action of potential drugs for the development of methods for the treatment of IS. The properties of some potential neuroprotective regulatory peptides are listed in Table 1.

## 2. Materials and Methods

In this review, we included articles and reviews published by PubMed until 6 March 2023. The queries used were “ischemic stroke and peptides”, “peptide regulation and transcriptomic analysis”, “neuroprotection and brain”, “ischemic stroke and neuroprotection”, and “peptide regulation and RNA-Seq.” The peptide sequence alignment was carried out using the MAFFT v7 tool [36]. BioRender software was used to produce scientific images and illustrations [37].

## 3. Results

### 3.1. Peptides Are New Potential Neuroprotective Agents for the Treatment of Acute IS

#### 3.1.1. Interfering Peptides

It is well known that protein–protein interactions (PPI) are involved in normal physiological processes. It is estimated that, in the human interactome, PPI networks include about 650,000 contacts [38]. To date, more than half a million forms of PPI dysregulation have been associated with pathological events, and addressing such dysregulation is considered a new therapeutic approach for the treatment of many diseases [17]. It has been found that short peptides that interfere with PPIs may have therapeutic effects. These peptides are called interfering peptides (IP). They are able to bind to the surfaces of proteins, thus blocking their interaction. Among the pathologies associated with PPI is IS.

As is known, excessive release of glutamate from synaptic endings and associated excitotoxicity is one of the main mechanisms causing neuronal death in stroke. A large number of studies have significantly expanded the understanding of the mechanisms underlying excitotoxicity in cerebral ischemia and identified many molecular targets for action, and thus, opened up new possibilities for neuroprotective strategies for ischemia [39,40,41,42]. The development of IPs that prevent excitotoxic stress during cerebral ischemia has become one of the strategies aimed at preventing neuronal death in the penumbra region surrounding the damage zone [43,44,45].

IPs include the recently developed synthetic R1-Pep (SETQDTMKTGSSTNNNEEEKSR) and PP2A-Pep (FQFTQNQKKEDSKTSTSV) peptides (Table 1), which inhibit the interaction of γ-aminobutyric acid type B (GABA_B_) receptors with enzymes involved in their phosphorylation [46,47]. Under physiological conditions, GABA_B_ receptors control the excitability of neurons in the brain through prolonged inhibition, and thereby counteract neuronal overexcitation and death. However, during cerebral ischemia, excitotoxic states rapidly downregulate GABA_B_ receptors through phosphorylation/dephosphorylation processes mediated by Ca^2+^/calmodulin-dependent protein kinase II (CaMKII) and protein phosphatase 2A (PP2A) [46,48,49]. After phosphorylation, GABA_B_ receptors undergo lysosomal degradation rather than being recycled back to the plasma membrane [48,50]. The R1-Pep peptide penetrating into cells inhibits the interaction of GABA_B_ receptors with CaMKII, preventing its phosphorylation. Administration of this peptide to cultured cortical neurons exposed to excitotoxic conditions, as well as its addition to mouse brain slices after middle cerebral artery occlusion (MCAO), restores the function of GABA_B_ receptors [46]. Another small interfering PP2A-Pep peptide, according to in vitro and ex vivo studies, inhibits the interaction of GABA_B_ receptors with PP2A and restores the expression of GABA_B_ receptors on the cell surface of neurons under normal physiological conditions and after excitotoxic stress [47]. The model of regulation for GABA_B_ receptor activity with the participation of R1-Pep and PP2A-Pep peptides is shown in Figure 1.

A current drug candidate for disrupting PPI in IS is the cell-penetrating peptide nerinetide (NA-1, YGRKKRRQRRRKLSSIESDV) (Table 1). The peptide consists of nine C-terminal residues of the NR2B subunit of the N-methyl-D-aspartate (NMDA) receptor, which selectively binds glutamate and aspartate. It prevents the post-synaptic density protein 95 (PSD-95) from binding to the NR2B subunit of the NMDA receptor and the PDZ domain of neuronal nitric oxide synthase (nNOS) [8,51]. Disruption of this PPI by NA-1 attenuates neurotoxic signaling cascades that lead to excessive calcium ion entry into neurons. NA-1 also prevents excitotoxic cell death and subsequent brain damage. Long-term evaluation of stroke outcomes in rodent and primate models of focal ischemia has also shown that NA-1 can reduce infarct size, along with an improvement in the neurological deficit. However, clinical studies have shown that treatment with a single dose of NA-1 along with endovascular thrombectomy did not improve long-term stroke outcome compared to patients treated with placebo [52]. At the same time, the cohort of patients treated with NA-1 without any thrombolytics had a lower risk of mortality, a significant reduction in infarct volume, and an improvement in functional outcome. This finding will require confirmation but suggests that neuroprotection in human stroke might be possible [52].

Thus, according to the data of in vitro and ex vivo studies, the developed interfering peptides exerted neuroprotective activity and inhibited excitotoxic neuronal death. IPs are believed to have great therapeutic potential for inhibiting progressive neuronal death in patients with acute stroke.

#### 3.1.2. Arginine-Rich Peptides

A relatively new class of compounds with a combination of different properties that provide a neuroprotective effect in the treatment of IS is cationic arginine-rich peptides (CARPs). Polyarginine peptides show high levels of cellular internalization and have high therapeutic potential. Guanidine groups in arginines form bidentate hydrogen bonds with negatively charged carboxyl, sulfate, and phosphate groups of proteins, mucopolysaccharides, and cell membrane phospholipids. Such interactions lead to the internalization of peptides into the cell under physiological conditions [53]. Possessing high anti-excitotoxic and anti-inflammatory neuroprotective efficacy, they are able to interfere with calcium influx into cells, stabilize mitochondria, inhibit proteolytic enzymes, induce survival signaling, and reduce oxidative stress [54,55,56,57]. Meloni et al. are actively studying peptides of this class in vitro for the activity of the thrombolytic agents alteplase (tPA) and tenecteplase (TNK). In one of the latest studies by the Meloni team, it was shown that arginine-rich peptides R18D (polyarginine-18, D-isomer) and R18 (polyarginine-18, L-isomer) (Table 1), when co-administered with thrombolytic agents, increase the maximum activity of the thrombolysis reaction while maintaining these neuroprotective properties [58]. Thus, the polyarginine peptides R18D and R18 represent new potential neuroprotective agents for the treatment of acute IS, the administration of which can have a significant effect in clinical use during clot thrombolysis.

Another polyarginine ST2-104 (ARSRLAELRGVPRGL) peptide (Table 1), obtained by the fusion of nona-arginine (R9) with a short peptide aptamer CBD3 from the collapsin response mediator protein 2 (CRMP2), protected neuroblastoma cells SH-SY5Y from death after exposure to glutamate, limited excess calcium intake, blocked apoptosis and autophagy, reduced infarct volumes, and improved neurological scores in MCAO-treated rats [59]. The neuroprotective effect of ST2-104 was due to its effect on apoptosis and autophagy via the CaMKKβ/AMPK/mTOR signaling pathway.

#### 3.1.3. Shuttle Peptides to Provide Neuroprotection

Another new therapeutic strategy for acute IS is the use of peptides as carriers (shuttles) that ensure the permeability of drugs through the blood–brain barrier (BBB) [60,61]. It is well known that the BBB is a structural barrier that ensures the supply of nutrients to the brain, protects it from harmful substances, and, at the same time, effectively blocks the entry of most neuroprotective agents. A useful property of peptides is their ability to overcome the BBB, which allows them to be used as a carrier for drug delivery. Brain-permeable peptide–drug conjugates, consisting of BBB shuttle peptides, linkers, and drug molecules, directly cross the BBB via an adsorption-mediated transcytosis pathway or through interaction with receptors or other proteins on the surfaces of BBB cells to initiate endogenous transcytosis or other means of transport [62,63,64].

It is well known that glycine has a neuroprotective effect in cerebral IR but has a low permeability through the BBB. A tripeptide, H-Gly-Cys-Phe-OH (GCF) (Table 1), which can permeate through the BBB, acts as a BBB shuttle and prodrug, delivering the amino acid glycine to the brain to provide neuroprotection [65].

A well-known cellular antioxidant enzyme is superoxide dismutase (SOD). However, exogenous SOD cannot be used to protect tissues from oxidative damage due to the low permeability of the cell membrane. The recombinant CPP-SOD fusion protein, which combines the SOD protein with short cell-penetrating peptides (CPPs) (Table 1), can cross the BBB and alleviate severe oxidative damage in various brain tissues by scavenging reactive oxygen species, reducing the expression of inflammatory factors, and inhibiting NF-κB/MAPK signaling pathways [61]. Thus, the clinical application of CPP-SOD impacts the damage associated with oxidative stress and new therapeutic strategies. Based on their physical and chemical properties, CPPs are classified as cationic, amphipathic, and hydrophobic peptides. CPPs typically contain more than five positively charged amino acids. Most cationic CPPs are derived from natural TAT (YGRKKRRQRRR) and penetratin (RQIKIWFQNRRMKWKK) peptides [53,66].

#### 3.1.4. Peptides That Mimic Natural Regulatory Peptides and Hormones

Many regulatory peptides are effective neuroprotectors. The main features of regulatory peptides are polyfunctionality, formation caused by cleavage from a precursor polypeptide, and a cascade mechanism of action. Neuropeptides include regulatory peptides that have a pleiotropic effect and are produced by neurons. The role of neuropeptides in the development of diseases and the treatment of neurological disorders was recently described in detail in a review by Yeo et al. [35].

New neuroprotective peptides with the potential to improve stroke outcomes include synthetic peptides that mimic natural regulatory peptides and hormones. They have greater metabolic stability, a longer half-life of their elimination from the blood, and a smaller size [35]. Synthetic adropin has therapeutic potential [67]. Adropin is a unique hormone encoded by the energy homeostasis-associated (*Enho*) gene [68]. This highly conserved peptide has many functions, including maintaining the integrity of the BBB and reducing the activity of matrix metalloproteinase-9 (MMP-9) [69]. The synthetic adropine (adropin(34–76), ACHSRSADVDSLSESSPNSSPGPCPEKAPPPQKPSHEGSYLLQP) treatment of IS mice reduced infarct size by activating eNOS and reducing oxidative damage [67] (Table 1). A recent study showed that in aged mice undergoing transient MCAO, post-ischemic therapy with synthetic adropine markedly reduced infarct volume, cerebral edema, and BBB damage; lowered MMP-9 levels; and significantly improved motor function, muscle strength, and long-term cognitive function [69]. 

The therapeutic effect in rats with MCAO was provided by another synthetic dynorphin A(1–8) peptide [70] (Table 1). This peptide is a fragment of dynorphin A with strong opioid activity. Dynorphin A(1–8) (YGGFLRRI) has been shown to inhibit oxidative stress and apoptosis in MCAO rats, affording neuroprotection through NMDA receptor and κ-opioid receptor channels [70]. In MCAO-treated rats, intranasal administration of dynorphin A(1–8) showed better behavioral improvement, a higher extent of Bcl-2 and activity of SOD, along with a much lower level of infarction volume, brain water content, number of cell apoptosis, extent of Bax, and Caspase-3 compared to the control [70]. 

The NX210 (WSGWSSCSRSCG) peptide is also a potential therapeutic agent against cerebral IR injury (Table 1). The peptide is derived from the thrombospondin type 1 repeat (TSR) sequence of SCO-spondin. NX210 prevents oxidative stress and neuronal apoptosis in cerebral IR through enhancement of the integrin-β1/PI3K/Akt signaling pathway [71].

At present, in the development of neuroprotective drugs, considerable attention has been paid to Glucagon-Like Peptide-1 (GLP-1, HAEGTFTSDVSSYLEGQAAKEFIAWLVKGRG) analogs used to treat type 2 diabetes and obesity. Diabetes is an important risk factor for cerebral infarction. In diabetic patients, the frequency of cerebral infarction is 1.8–6.0 times higher compared to non-diabetic patients [72]. One of the main biological targets for the pharmaceutical action of the pleiotropic GLP-1 hormone and its analogues is the glucagon-like peptide-1 receptor (GLP-1R). Binding to this receptor stimulates the secretion of insulin by pancreatic β-cells, thereby causing a decrease in glycemia. There are currently seven GLP-1R agonists (GL1-RAs) approved for the treatment of diabetes [73]. At the same time, the long-acting synthetic GLP-1 analogues Liraglutide [74] and Semaglutide [75] demonstrated the most notable therapeutic success.

Liraglutide (HAEGTFTSDVSSYLEGQAAK(E-C16 fatty acid)EFIAWLVRGRG) is a 32 amino acid peptide with C-16 fatty acid fragments. It has 97% identity with the human GLP-1 sequence. Numerous studies using models of cerebral ischemia in rodents indicate that Liraglutide reduces the volume of the infarct zone, has a neuroprotective and antioxidant effect, promotes angiogenesis and increases the expression of VEGF in the area of cerebral ischemia [76,77,78], improves metabolic and functional recovery after stroke [79], and reduces neurological deficits [80,81,82].

The study of the mechanism of action of Liraglutide in the model of rat cerebral IR with diabetes showed that the peptide inhibited endoplasmic reticulum stress and thereby reduced apoptosis [82]. A recent study by Yang et al. disclosed a new neuroprotective mechanism by which Liraglutide provides protection against damage caused by cerebral ischemia [83]. Using a mouse model of focal ischemia of the cerebral cortex and microglial cells, the authors showed that the neuroprotective effect of Liraglutide can be achieved by inhibiting pyroptosis, an inflammatory form of programmed cell death. In this case, the anti-pyroptotic mechanism of Liraglutide in vivo can be mediated by NOD-like receptor protein 3 (NLRP3) [83].

Semaglutide (HXEGTFTSDVSSYLEGQAAK(C18diacid-γE-OEG-OEG)EFIAWLVRGRG), another long acting GL1-RA, has two amino acid substitutions compared to human GLP-1 (2-aminoisobutyric acid (Aib) 8, arginine (Arg) 34) and is derivatized at lysine 26 [84]. Subcutaneous Semaglutide, administered once a week, was first approved for permanent weight management in June 2021 in the United States. Recent studies indicate that Semaglutide is not only safe and effective in the treatment of obesity [85], but also reduces ischemic cerebrovascular events in type 2 diabetes [86]. Currently, in overweight or obese patients, the protective effect of Semaglutide in heart disease and stroke is being actively studied [87,88].

Thus, according to numerous studies, GL1-R agonist drugs recommended for the treatment of diabetes and obesity exhibit a neuroprotective effect and provide protection against damage caused by cerebral ischemia [76,77,78,79,80,81,82,83,86,87,88]. It can be assumed that the effects of these neuropeptides are much wider than currently recognized. We believe that GL1-R agonists are a potential therapeutic tool to protect the brain during strokes.

In recent years, drugs based on melanocortin peptides have been actively developed. Melanocortins are a large family of neuropeptides formed from a common precursor, the proopiomelanocortin molecule, which includes the adrenocorticotropic hormone (ACTH), a group (α-, β-, γ-) of melanocyte-stimulating hormones (MSH). Melanocortins have a wide spectrum of physiological activity, which makes it possible to use their fragments for drug development. The nootropic peptide Semax (MEHFPGP) is successfully used for the treatment of IS (Table 1). The N-terminus of Semax contains an ACTH(4–7) fragment, and the C-terminus is stabilized by the addition of the tripeptide Pro-Gly-Pro (PGP). Semax has been used in neurological practice for many years in the treatment of acute and chronic disorders, including IS and its consequences [89,90]. Semax has a pronounced nootropic, neuroprotective, and immunomodulatory effect [91,92]. 

Recent studies have shown that another melanocortin ACTH(6-9)PGP (HFRWPGP) peptide has a wide spectrum of neuroprotective activity [93,94,95]. A study of the effect of the ACTH(6–9)PGP peptide on the survival of cultured cortical neurons against the background of the excitotoxic effect of glutamate showed that, depending on the dose, the peptide protected neurons from cell death [94] (Table 1). The neuroprotective effect of ACTH(6–9)PGP was accompanied by a slowdown in the development of delayed calcium dysregulation and synchronous depolarization of mitochondria. The peptide significantly increased the number of neurons that restored calcium ion homeostasis after glutamate withdrawal. A subsequent study of the proliferative and cytoprotective activity of ACTH(6–9)PGP peptide on SH-SY5Y cells in models of toxicity caused by hydrogen peroxide, tert-butyl hydroperoxide, or potassium cyanide (KCN) showed that the peptide dose-dependently protected cells from oxidative stress and exhibited proliferative activity. The mechanism of peptide action was the modulation of proliferation-related NF-κB genes and stimulation of the pro-survival NRF2-gene-related pathway, as well as a decrease in apoptosis [95]. Since reperfusion used to treat IS causes additional damage in brain cells, including the accumulation of excess oxygen radicals and activation of apoptosis, the discovered therapeutic effects of ACTH(6–9)PGP allow us to highly appreciate the possibility of its clinical use after the administration of thrombolytics (e.g., tPA) or mechanical thrombectomy.

**Table 1 genes-14-00953-t001:** New potential neuroprotective peptides and their functions.

Type of Peptide	Peptides	Functions	References
Interfering peptides (IPs)	R1-Pep	It inhibits the interaction of GABA_B_ receptor with CaMKII, preventing receptor phosphorylation.	[46]
	PP2A-Pep	It inhibits the interaction of GABA_B_ receptor with PP2A, preventing receptor dephosphorylation.	[47]
	NA-1	It attenuates neurotoxic signaling cascades that lead to excessive calcium ion entry into neurons.	[8,51]
Cationic arginine-rich peptides (CARPs)	R18D, R18	They have high anti-excitotoxic and anti-inflammatory efficiency, and are able to interfere with calcium influx into cells, stabilize mitochondria, inhibit proteolytic enzymes, induce survival signaling, and reduce oxidative stress. Peptides increase the maximum activity of the thrombolysis reaction when co-administered with thrombolytic agents	[54,55,56,57,58]
	ST2-104	It effects on apoptosis and autophagy via the CaMKKβ/AMPK/mTOR signaling pathway.	[59]
Shuttle peptides	GCF	It acts as a BBB shuttle and prodrug, delivering the amino acid glycine to the brain to provide neuroprotection.	[65]
	CPP-SOD	The recombinant CPP-SOD fusion protein can cross the BBB and alleviate severe oxidative damage in various brain tissues by scavenging reactive oxygen species, reducing the expression of inflammatory factors and inhibiting NF-κB/MAPK signaling pathways.	[61]
Peptides that mimic natural regulatory peptides and hormones	Adropin(34–76)	It reduces infarct size by activating eNOS and reducing oxidative damage, maintains the integrity of the BBB and reduces the activity of MMP-9.	[67,69]
	Dynorphin A(1–8)	It affords neuroprotection through NMDA receptor and κ-opioid receptor channels.	[70]
	NX210	It prevents oxidative stress and neuronal apoptosis in cerebral IR through enhancement of the integrin-β1/PI3K/Akt signaling pathway.	[71]
	Liraglutide	Long acting GL1-RA that promotes angiogenesis, reduces neurological deficits, apoptosis, inhibits pyroptosis, an inflammatory form of programmed cell death.	[74,76,77,78,79,80,81,82,83]
	Semaglutide	Long acting GL1-RA that reduces ischemic cerebrovascular events in type 2 diabetes	[75,85,86,87,88]
	Semax	It has a pronounced nootropic, neuroprotective, and immunomodulatory effects. Peptide initiates a neurotransmitter and anti-inflammatory response.	[27,91,92,96,97]
	ACTH(6–9)PGP	It protects neurons from cell death, protected cells from oxidative stress and exhibited proliferative activity.	[94,95]

It should be noted that peptides often combine a few units to have different properties. In this review, we note a case where ACTH fragments were stabilized by the glyproline PGP unit. As a result, the Semax and ACTH(6-9)PGP peptides were created [89,90,93,94,95]. Furthermore, the N-terminus of R1-Pep and PP2A-Pep IPs were conjugated with a peptide sequence derived from the rabies virus glycoprotein (YTIWMPENPRPGTPCDIFTNSRGKRASNGGGG) to make the IPs penetrate cell membranes. In addition, there are examples of R1-Pep penetration being increased via conjugation with CARPs or co-conjugation with CARPs and CPPs (e.g. YTIWMPENPRPGTPCDIFTNSRGKRASNGGGG-RRRRRRRRR-SETQDTMKTGSSTNNNEEEKSR) [46,47]. Furthermore, GL1-RAs, including liraglutide and semaglutide, contain additional lateral fatty rather than amino acid sequences, which improve pharmacokinetics and protect the peptide from both peptidase degradation and renal filtration [98].

Figure 2 shows multiple sequence alignments for IPs, CARPs, CPPs, and mimics of natural regulatory peptides described in this review. The results were obtained using the MAFFT v7 tool. The similarity between the structures of the GLP-1-related peptides (GLP-1, major branches of liraglutide and semaglutide) is visible. In addition, arginine-containing peptides (TAT, R18, R18D) clustered. Moreover, the NA-1 peptide, which is related to the IP group, belonged to this cluster too. Subsequently, PGP-containing peptides (PGP, Semax, ACTH(6–9)PGP, and adropin(34–76)) clustered independently. Concomitantly, the IPs did not cluster. Thus, R1-Pep, PP2A-Pep, and NA-1 did not have similar domains based on the results obtained from the MAFFT v7 tool. Interestingly, the dynorphin A(1–8), NX210, ST2-104, and GCF peptides did not belong to any cluster. Perhaps peptides have more complex and nonlinear correlations between their structures and neuroprotective properties.

### 3.2. Transcriptomic Analysis as a New Approach to Reveal the Molecular Mechanisms of Ischemic Damage and the Action of Potential Neuroprotectors

At present, transcriptomics has become an effective approach for studying the mechanisms of pathological processes in various diseases and searching for molecular targets for their drug treatment. High-throughput mRNA sequencing (RNA-Seq) reveals information about the expression of individual genes and makes it possible to identify signaling pathways involved in the development of many diseases. Transcriptomic analysis has made a significant contribution to the study of the molecular mechanisms of brain damage as a result of IR [99,100,101]. This approach has also made it possible to reveal the mechanisms of therapeutic effects of many potential drugs at the genetic level, creating a theoretical basis for the treatment of IS [102,103,104,105,106].

RNA-Seq analysis revealed the molecular mechanisms by which regulatory peptides and peptide-related drugs perform a neuroprotective role in cerebral ischemia. There are many examples of the use of transcriptome analysis to study the mechanisms of action of a number of peptides, including Orexin-A [107], Semax [27], VR-10 [29], and semaglutide [101].

Recently, using RNA-Seq, we studied the protective properties of the Semax peptide at the transcriptome level under transient MCAO conditions. Previously, we studied the gene expression in rat brain under conditions of incomplete global ischemia of the rat brain and permanent MCAO using real-time reverse transcription polymerase chain reaction (RT-PCR). Thereby, we showed that the peptide affects the expression of a limited number of genes encoding neurotrophic factors and their receptors [108,109]. Using RatRef-12 BeadChips, it was shown that Semax affects the expression of genes associated with the immune and vascular systems in the brain of rats after permanent MCAO [92,97]. Using RNA-Seq analysis, we identified several hundred differentially expressed genes (DEGs) (>1.5-fold change) in the brains of rats 24 h after transient MCAO treated with Semax compared with control animals treated with saline [27,110]. We found that Semax suppressed the expression of genes associated with inflammatory processes (e.g., *Hspb1, Fos, iL1b, iL6, Ccl3, Socs3*) and activated the expression of genes associated with neurotransmission (e.g., *Cplx2, Chrm1, Gabra5, Gria3, Neurod6, Ptk2b*), while IR, on the contrary, activated the expression of genes involved in inflammation, immune response, apoptosis, and stress response and suppressed the expression of genes associated with neurotransmission. Analysis of signaling pathways associated with Semax-induced DEGs in the transient MCAO rat model using a web server for functional enrichment analysis (g:ProfileR) and Gene Set Enrichment Analysis (GSEA) data showed that DEG activation in Semax-treated rats 24 h after MCAO was associated with calcium signaling, dopaminergic, cholinergic, the glutamatergic synapse, and G-protein coupled receptor (GPCR) signaling through chemical synapses, whereas DEG suppression was associated with the phagosome, interleukin 17 (IL-17), tumor necrosis factor (TNF), p53 signaling pathways, innate immune system, neutrophil degranulation, and cytokine signaling in the immune system [27]. Thus, according to RNA-Seq data, 24 h after transient MCAO, Semax initiated a neurotransmitter and anti-inflammatory response that compensated for mRNA expression patterns that were disturbed under IR conditions. Perhaps, Semax can mediate its influence on gene expression indirectly through interaction with receptors on the cell membrane. Moreover, the action of Semax, which is able to reduce the disturbances caused by ischemia, can be explained by joint action of peptide on receptors in an allosteric manner, together with hormones and mediators of the ischemic response. The model of the influence of Semax on the transcriptome of brain cells and the spectrum of Semax effects identified at the gene expression level at 24 h after transient MCAO in rats is shown in Figure 3.

Semax, as noted above, contains an ACTH(4–7) fragment flanked at the C-terminus by the PGP tripeptide. There is evidence that PGP not only ensures the resistance of peptides against biodegradation, but also exhibits a wide range of biological activity. Namely, PGP is involved in the formation of the immune response, and the PGP peptide has antiplatelet, anticoagulant, fibrinolytic, anxiolytic, antiapoptotic, and antistress activity [108,111,112,113,114,115,116]. Using real-time RT-PCR, we studied the effects of PGP and another PGP-containing the Pro-Gly-Pro-Leu (PGPL) peptide on the expression of a number of inflammatory cluster (IC) and neurotransmitter cluster (NC) genes in rat brain under transient MCAO. Then, we compared the studied peptide effects with the action of Semax under IR conditions [96,117]. Both PGP and PGPL peptides showed an effect dissimilar to the effects of Semax at 24 h after transient MCAO. The administration of peptides did not have a statistically significant effect on the expression of genes involved in inflammation. This result highlights the importance of the structure of the ACTH(4–7) fragment for the effects of Semax. In addition, the IC (*iL1b*, *iL6*, and *Socs3*) rat genes for PGP, as well as the IC (*iL6, Ccl3, Socs3,* and *Fos*) and NC (*Cplx2, Neurod6,* and *Ptk2b*) rat genes for PGPL were discussed. The expression levels of these genes changed significantly after the corresponding peptide, compared to Semax administration. Based on the results of the analysis of gene expression under experimental conditions using bioinformatic approaches, a functional network was built that illustrates the spectra of common and unique effects of PGP, PGPL, and Semax peptides [117]. Thus, transcriptomic analysis, in addition to studying the molecular mechanisms of action of peptides, makes it possible to reveal the relationship between their chemical structure and possible effects at the genomic and functional levels.

## 4. Discussion

The progress achieved in the recovery of patients after IS using pharmacological thrombolysis and mechanical thrombectomy has not solved the problem of limiting progressive neuronal death after stroke. There is still a need for further search for therapeutic agents for the treatment of stroke. As can be seen from the available literature, a large number of studies using oxygen-glucose deprivation and reperfusion (OGD/R) cell cultures, as well as laboratory animal models of experimental brain ischemia mimicking conditions of ischemic injury, have demonstrated the potential benefit of neuroprotection. Neuroprotective peptides could be the answer to medical demands in the treatment of strokes and their consequences. To date, hundreds of peptides have undergone preclinical or clinical studies [8]. As noted, peptides are an extensive class of molecules with a variety of structures and functions. In our review, we have identified several groups of peptides that can exhibit valuable properties for neuroprotection. Thus, IPs affect the activity of receptor systems, blocking PPIs, and the effects of IPs may be important in overcoming the neurotoxic effects of ischemic injuries. Peptides that mimic natural regulatory peptides and hormones serve as a basis for creating drugs. As a result, a GLP-1 analog (semaglutide), as well as an ACTH analog (Semax), will potentially be used as medicines in different countries [89,118]. The mechanisms of action for CARPs are less studied, but they have multiple antitoxic and anti-inflammatory effects during ischemia, affecting the activity of numerous signaling systems. It should be noted that the search for targeted drug delivery methods while observing the principles of safety and efficacy in therapy is one of the most relevant current topics. In this regard, shuttle peptides, which are carriers of biologically active molecules across the BBB, are of interest. The complexes of CPPs and proteins (for example, CPP-SOD) ensure the efficient penetration of the enzyme into the target area of the cell in realizing the therapeutic effect. Moreover, many peptides combine two valuable qualities at the same time. Examples of such convergent actions are peptides formed by the fusion of several peptides with different properties. Thus, the Semax peptide has a significant neuroprotective effect in stroke, including after intraperitoneal administration. Semax contains a PGP residue that increases metabolic resistance. It is important to note that peptides might have more complex and nonlinear correlations between their structures and neuroprotective properties. Thus, many peptides belonging to the same group do not structurally repeat each other, as per the results we obtained using MAFFT v7.

From our point of view, peptides have another fundamentally important property. Peptides have multiple (pleiotropic) effects on receptor systems [119,120,121]. This property of peptides is the key when treating multifactorial diseases with a complex cascade of events, including strokes. Acute ischemia leads to multiple membrane polarizations and the activation of brain-cell receptors due to various factors caused by ischemic damage. This, in turn, triggers various signaling pathways that mediate the influence of many gene transcriptions inside the cell nucleus [28,110,122,123,124]. Peptides can allosterically interact with receptors and modulate the multiple signals that cells receive during a stroke [125,126]. It is also assumed that the pleiotropic properties of peptides allow for the formation of neuroprotective polyfunctional responses to overcome multiple reactive damage processes. The involvement of peptides in numerous network interactions, including multiple allosteric modulations of receptor activities, requires specific approaches to elucidate their mechanisms of action.

The current development of omics technology can be especially valuable in this regard. Omics technologies make it possible to obtain multiple genome-wide data arrays at a time and carry out multiple results comparisons in the conditions of one experiment. Studies based on transcriptomic and proteomic approaches can reveal the expression profiles of RNAs (RNA-Seq, single-cell RNA-Seq), proteins, and peptides to assess how much neuropathology has distorted biomolecule levels [122,124,127,128,129,130,131]. At the same time, the normalization (compensation) of the disturbed profiles of the analyzed molecules after exposure to the peptide is evidence of the potential drug (peptide)’s neuroprotective properties under neuropathological conditions. One stage of testing peptides for neuroprotective properties can be based on this fact. Further applications of the functional enrichment and data clustering methods make it possible to elucidate the metabolic systems the peptide is involved in modulating. Potentially, with the accumulation of a lot of omics data, machine learning and artificial intelligence methods may reveal the relationship between peptide structures and their functions. Then, the prediction of peptide structures required for drug properties can be realized. Thus, the frontiers of peptide drug development can be expanded by converging natural, health, and computer sciences.

## 5. Conclusions

It can be assumed that the latest developments in the creation of neuroprotective agents, combined with new possibilities for their delivery to the brain, an understanding of the pathogenesis of stroke in general, and consideration of the previous shortcomings in preclinical and clinical studies, will provide a breakthrough in the treatment of IS. Here, we reviewed the most recent strategies to search for neuroprotectors for the treatment of IS, advances in the development of biologically active neuroprotective peptides, and the role of genome-wide transcriptome studies in identifying the molecular mechanisms of action of potential drugs.

## Figures and Tables

**Figure 1 genes-14-00953-f001:**
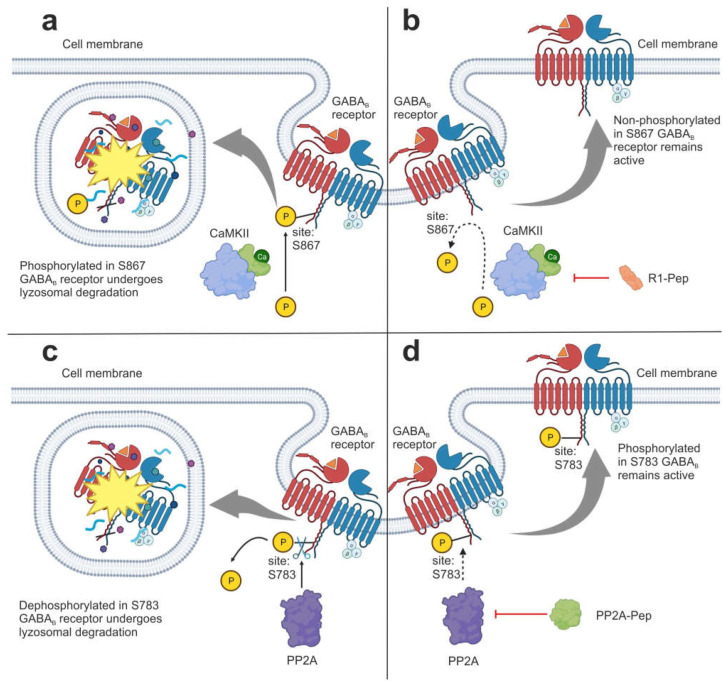
Effect of synthetic inhibitors on GABA_B_ receptor recirculation. (**a)** Phosphorylation of GABA_B_ in the S867 site mediated by Ca^2+^/calmodulin-dependent protein kinase II (CaMKII) leads to lysosomal degradation in the receptor. (**b**) The R1-Pep peptide inhibits the GABA_B_ receptors’ interactions with CaMKII, preventing phosphorylation. Non-phosphorylated S867 receptors tend to return to the cell surface instead of being degraded. (**c**) Dephosphorylation of GABA_B_ receptors in the S783 site mediated by protein phosphatase 2A (PP2A) leads to lysosomal degradation in the receptor. (**d**) PP2A-Pep inhibits the GABA_B_ receptor’s interaction with PP2A, preventing its phosphorylation. Receptors with phosphate in the S783 site tend to return to the cell surface instead of being degraded. The illustration was made using BioRender.

**Figure 2 genes-14-00953-f002:**
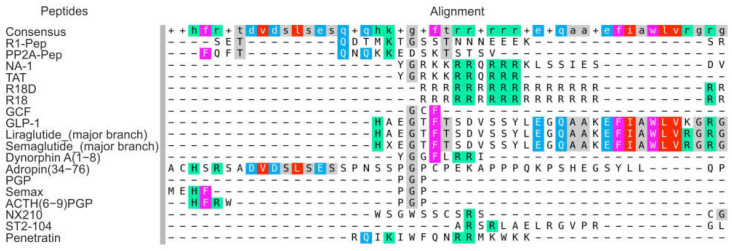
The sequence alignment of the IPs, CARPs, CPPs, and mimics of natural regulatory peptides described in the review. The results were obtained using the MAFFT v7 tool.

**Figure 3 genes-14-00953-f003:**
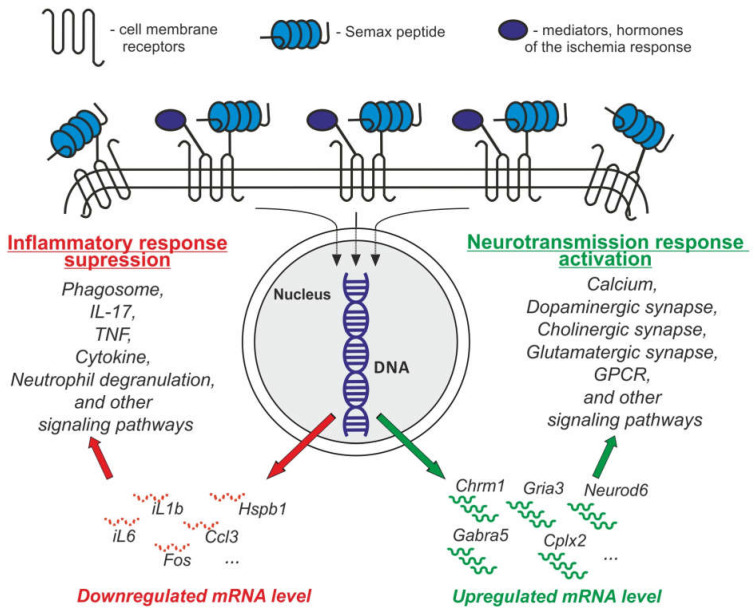
The model of the influence of Semax on the transcriptome of brain cells, and the spectrum of Semax effects identified at the gene expression level at 24 h after transient MCAO in rats.

## Data Availability

Data sharing not applicable.

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
