# Peer review of "Neuroprotective Peptides and New Strategies for Ischemic Stroke Drug Discoveries"

_genes, 2023, doi:10.3390/genes14050953_

Round 1

Reviewer 1 Report

The main question that is addressed by Dergunova et al. is to highlight the drawbacks of conventional therapies to approach stroke and introduce peptide therapy instead. The topic is interesting for the reader but does not seem to be inclusive. The whole manuscript is observed in Table 1 and can be published as a short communication. Although they have mentioned the keywords in the materials and methods, they should describe, classify, and cite enough literature on neuro-protectors and then focus on neuroprotectors in stroke. There are several other comments which are mandatory to be clearly and concisely described as follows:

Comment 1: The manuscript needs several schematics or figures for various sections. For example, the cascade which is described in the last paragraph of page 2 and the first paragraph of page 3 regarding interference of R1-PEP and PP2A-PEP with GABA receptors must be drawn. This is similar for nerinetide and its mechanisms of action.

Comment 2: There is no sequence and structural data available for peptides (except for Semax) in the text or table. Authors should provide this data somewhere in the manuscript (at least sequences for short peptides) or provide UniProt ID for long peptides. What are the physiochemical characteristics of each category? The Arg-rich peptides seem to be cationic. What about other groups and examples that are provided in the manuscript? Do you observe any amphipathicity or hydrophobicity? How can an interacting peptide be differentiated from a shuttle peptide regarding structure? What is their structure when they are in an aqueous environment compared to membrane-like environments?

Comment 3: In line with the previous comment, authors should describe how the encrypted information in the “omics” data is interpreted as to be a “neuroprotective peptide” to design further experimental verification. Do new unknown neuropeptides show similarities to other known neuropeptides using blast? Is there any conserved motif?

Comment 4: The section which is provided as a discussion does not add any value to the manuscript and has complete redundancy with the introduction.

Comment 5: The conclusion is not satisfactory and needs to be rewritten.

Author Response

Response to the comments of Reviewer 1 to Manuscript ID: genes-2264311

Authors:

We are very grateful to the Reviewer 1 for the review and constructive comments. We carefully considered the comments of the Reviewer 1 and attached the answers to all comments.

Reviewer 1:

  1. Although they have mentioned the keywords in the materials and methods, they should describe, classify, and cite enough literature on neuro-protectors and then focus on neuroprotectors in stroke.

Authors:

In accordance with the Reviewer’s recommendation, changes were added in the text (lines 39-53, 57, 58, 73-75, 77-79, 87-90 in Mark-up copy).

Reviewer 1:

  1. The manuscript needs several schematics or figures for various sections. For example, the cascade which is described in the last paragraph of page 2 and the first paragraph of page 3 regarding interference of R1-PEP and PP2A-PEP with GABA receptors must be drawn. This is similar for nerinetide and its mechanisms of action.

Authors:

In accordance with the Reviewer’s recommendation, Figure was added (lines 91, 128-130, 152-162 in Mark-up copy).

Reviewer 1:                                                                                       

  1. There is no sequence and structural data available for peptides (except for Semax) in the text or table. Authors should provide this data somewhere in the manuscript (at least sequences for short peptides) or provide UniProt ID for long peptides. What are the physiochemical characteristics of each category? The Arg-rich peptides seem to be cationic. What about other groups and examples that are provided in the manuscript? Do you observe any amphipathicity or hydrophobicity? How can an interacting peptide be differentiated from a shuttle peptide regarding structure? What is their structure when they are in an aqueous environment compared to membrane-like environments?

Authors:

In accordance with the Reviewer’s recommendation, changes were added in the text (lines 111, 112, 133, 182, 212-216, 232, 233, 241, 248, 254, 255, 264, 280, 281, 302, 308, 330-341 in Mark-up copy).

Reviewer 1:

  1. In line with the previous comment, authors should describe how the encrypted information in the “omics” data is interpreted as to be a “neuroprotective peptide” to design further experimental verification. Do new unknown neuropeptides show similarities to other known neuropeptides using blast? Is there any conserved motif?

Authors:

In accordance with the Reviewer’s recommendation, changes were made (lines 90, 91, 342-356, 466-469, 483-500 in Mark-up copy). Peptide multiple sequence alignment using MAFFT v7 has been added in Figure 2.

Reviewer 1:

  1. The section which is provided as a discussion does not add any value to the manuscript and has complete redundancy with the introduction.

Authors:

In accordance with the Reviewer’s recommendation, changes were made (lines 442-511 in Mark-up copy).

Reviewer 1:

  1. The conclusion is not satisfactory and needs to be rewritten.

Authors:

In accordance with the Reviewer’s recommendation, changes were made (lines 513-516 in Mark-up copy).

Reviewer 2 Report

1) Title does not reflect the contents of a review article. The title refers to “Regulatory Peptides” but article fails to address any regulatory peptides.

The authors fail to mention very well known neuropeptides including various platelet-derived growth factors, various calcitonin peptides, and sectroneurin. In addition, a recent review, “Potentials of Neuropeptides as Therapeutic Agents for Neurological Diseases” (https://pubmed.ncbi.nlm.nih.gov/35203552/) that covers many important regulatory peptides relevant to ischemic stroke.

2) As a review article the authors must include an instructional paragraph on ‘regulatory peptides’ including general definition and roles in ischemic stroke.

3) Materials and Methods requires more details. As required by Genes: “you must make all materials, data, and protocols associated with the publication available to readers.” Therefore, the actual complete queries must be included or at least the most useful queries used.

Also, the authors must also use all variation of ‘next generation sequencing’ not just RNA-seq.

Specific comments by line:

Line 175: Remove the word ‘Natural’.

Lines 175 and 176: Is adrophin a hormone or a peptide? Be consistent because not all hormones are peptides and not all peptides are hormones.

Line 184: dynorphin A(18) is not a synthetic peptide (known since 1998). Also correct this in Table 1.

Lines 196 through 229: Remove or reduced and rewrite because these do not refer to “ischemic stroke” nor provide “neuroprotection”. Diabetes and obesity are high risk factors for stroke but neither are direct or specific cause of ischemic strokes. So peptides treating diabetes and obesity do not directly provide neuroprotection.

Line 235: define MSH.

Line 237: Remove the word “neuropeptide because Semax is an synthetic construct.

Lines 292, 293, 328, 329: Gene names are not consistent. Capitalize the symbols and remove italics for gene names or italicize all gene names. Why is Hspb1 underlined?

Line 297: Define the “tMCAO model” and “gProfileR and GSEAS data”.

Line 300; Remove ‘etc’ and end the sentence.

Lines 300-302; Define IL-17 and TNF.

Line 302; Remove ‘etc’.

Line 334: Provide the names of three peptides.

Author Response

Response to the comments of Reviewer 2 to Manuscript ID: genes-2264311

Authors:

We are very grateful to the Reviewer 2 for the review and constructive comments. We carefully considered the comments of the Reviewer 2 and attached the answers to all comments.

Reviewer 2:

  1. Title does not reflect the contents of a review article. The title refers to “Regulatory Peptides” but article fails to address any regulatory peptides.

Authors:

In accordance with the Reviewer’s recommendation, the Title was changed (lines 2-4 in Mark-up copy).

Reviewer 2:

  • The authors fail to mention very well known neuropeptides including various platelet-derived growth factors, various calcitonin peptides, and sectroneurin. In addition, a recent review, “Potentials of Neuropeptides as Therapeutic Agents for Neurological Diseases” (https://pubmed.ncbi.nlm.nih.gov/35203552/) that covers many important regulatory peptides relevant to ischemic stroke.

Authors:

In accordance with the Reviewer’s recommendation, changes were made (lines 73-75, 77-79 in Mark-up copy).

Reviewer 2:

  1. As a review article the authors must include an instructional paragraph on ‘regulatory peptides’ including general definition and roles in ischemic stroke.

Authors:

In accordance with the Reviewer’s recommendation, changes were made (lines 219-224, 226-228 in Mark-up copy).

Reviewer 2:

  1. Materials and Methods requires more details. As required by Genes: “you must make all materials, data, and protocols associated with the publication available to readers.” Therefore, the actual complete queries must be included or at least the most useful queries used.

Authors:

In accordance with the Reviewer’s recommendation, changes were added (lines 87-91 in Mark-up copy).

Reviewer 2:

  • Also, the authors must also use all variation of ‘next generation sequencing’ not just RNA-seq.

In accordance with the Reviewer’s recommendation, other high-throughput methods including single-cell RNA-Seq and chromato-mass-spectrometric analysis were mentioned in the text. The changes were made (lines 486-488 in Mark-up copy).

Reviewer 2:

  • Line 175: Remove the word ‘Natural’.

Authors:

In accordance with the Reviewer’s recommendation, changes were made (line 228 in Mark-up copy).

Reviewer 2:

  • Lines 175 and 176: Is adrophin a hormone or a peptide? Be consistent because not all hormones are peptides and not all peptides are hormones.

Authors:

Adropin is both a hormone and a peptide (please see lines 228-230). The sequences of adropin(34-76) peptide was added (lines 232-233 in Mark-up copy).

Reviewer 2:

  • Line 184: dynorphin A(18) is not a synthetic peptide (known since 1998). Also correct this in Table 1.

Authors:

In accordance with the Reviewer’s recommendation, changes were made (lines 218, 328 in Mark-up copy).

Reviewer 2:

  • Lines 196 through 229: Remove or reduced and rewrite because these do not refer to “ischemic stroke” nor provide “neuroprotection”. Diabetes and obesity are high risk factors for stroke but neither are direct or specific cause of ischemic strokes. So peptides treating diabetes and obesity do not directly provide neuroprotection.

Authors:

We are very grateful to the Reviewer 2 for your comments. According to numerous studies, GL1-R agonist drugs recommended for the treatment of diabetes and obesity exhibit a neuroprotective effect and provide protection against damage caused by cerebral ischemia (Bai et al. 2021; Banerjee et al. 2023; Briyal, Shah, and Gulati 2014; Chen et al. 2018; Deng et al. 2018; Dong et al. 2017; He et al. 2020; Lingvay et al. 2023; Mares, Chatterjee, and Mukherjee 2022; Sato et al. 2013; Yang et al. 2022). It can be assumed that the effects of these neuropeptides are much wider than currently recognized. We believe that GL1-R agonists are a potential therapeutic tool to protect the brain during strokes. In accordance with the Reviewer’s recommendation, changes were made (lines 290-295 in Mark-up copy).

Reviewer 2:

  • Line 235: define MSH.

Authors:

In accordance with the Reviewer’s recommendation, changes were made (line 299 in Mark-up copy).

Reviewer 2:

  • Line 237: Remove the word “neuropeptide because Semax is an synthetic construct.

Authors:

In accordance with the Reviewer’s recommendation, changes were made (line 301 in Mark-up copy).

Reviewer 2:

  • Lines 292, 293, 328, 329: Gene names are not consistent. Capitalize the symbols and remove italics for gene names or italicize all gene names. Why is Hspb1 underlined?

Authors:

In accordance with the Reviewer’s recommendation, changes were made (lines 385-387, 424, 425 in Mark-up copy). Rat gene symbols are usually written in italics and with a capital letter. On the contrary, italics are not used to written protein symbols.

Reviewer 2:

  • Line 297: Define the “tMCAO model” and “gProfileR and GSEAS data”.

Authors:

In accordance with the Reviewer’s recommendation, changes were made (lines 390-392, 406, 410 in Mark-up copy).

Reviewer 2:

  • Line 300; Remove ‘etc’ and end the sentence.

Authors:

In accordance with the Reviewer’s recommendation, changes were made (line 395 in Mark-up copy).

Reviewer 2:

  • Lines 300-302; Define IL-17 and TNF.

Authors:

In accordance with the Reviewer’s recommendation, changes were made (lines 395, 396 in Mark-up copy).

Reviewer 2:

  • Line 302; Remove ‘etc’.

Authors:

In accordance with the Reviewer’s recommendation, changes were made (line 397 in Mark-up copy).

Reviewer 2:

  • Line 334: Provide the names of three peptides.

Authors:

In accordance with the Reviewer’s recommendation, changes were made (line 430 in Mark-up copy).

Round 2

Reviewer 1 Report

It was expected that the response to each comment would be added to the cover letter not only the line number.